# Forest Bathing Increases Adolescents’ Mental Well-Being: A Mixed-Methods Study

**DOI:** 10.3390/ijerph21010008

**Published:** 2023-12-20

**Authors:** Jennifer Keller, Jean Kayira, Louise Chawla, Jason L. Rhoades

**Affiliations:** 1Department of Environmental Studies, Antioch University, New England, Keene, NH 03431, USA; jrhoades@antioch.edu; 2Department of Environmental Studies, State University of New York College of Environmental Science and Forestry, Suracuse, NY 13201, USA; jckayira@esf.edu; 3Program in Environmental Design, University of Colorado Boulder, Boulder, CO 80309, USA; louise.chawla@colorado.edu

**Keywords:** forest bathing, forest therapy, adolescent well being, ecoanxiety, YPAR, mixed methods, photovoice, nature connection, nature therapy, mental well being

## Abstract

Previous research has demonstrated that practicing forest bathing has significant positive effects on adult psychological well-being. Considering the ongoing adolescents’ mental health crisis of increasing anxiety and depression, determining whether forest bathing has similar effects on adolescents is an important expansion of forest bathing research. This study investigated the possibility that forest bathing could improve adolescents’ mental well-being and sought to determine participants’ experiences of forest bathing. It used a convergent, parallel, mixed-methods design that was partially co-created with 24 participants aged 16–18 as part of a youth participatory action research (YPAR) project in which participants practiced forest bathing three times over 3 weeks. As measured using the Warwick–Edinburgh Mental Well-Being Survey, the mean participant mental well-being increased significantly after forest bathing, with moderate to large effect sizes. Participants described reduced stress and increased feelings of relaxation, peace, and happiness. These findings correlate with previous forest bathing research involving adult participants. It is recommended that educators and others who work with adolescents consider forest bathing as a simple, low-cost way to improve adolescents’ mental well-being.

## 1. Introduction

“I firmly believe that nature brings solace in all troubles”.—Anne Frank

Anne Frank’s words remind us that spending time in nature has long been a source of rejuvenation and healing. Empirical evidence supports this observation, and many studies have shown a link between spending time in green spaces, such as forests, nature preserves, and parks, or blue spaces, such as outdoor water environments, and improved mental and physical well-being [1,2,3,4,5,6,7]. More recently, research has also supported the benefits of practicing forest bathing, a nature-based mindfulness practice [8]. Forest bathing, often called nature and forest therapy in North America, involves spending time in forested areas to invite healing interactions [9]. Forest bathing evolved from the Japanese practice of shinrin-yoku and is described as making contact with and taking in the atmosphere of a forest [10].

Practicing forest bathing has many research-backed benefits for its practitioners. Most studies of the impacts of practicing forest bathing on human physiology have been completed in Japan and South Korea, where shinrin-yoku is considered part of preventive medicine [8]. Much of the research on the physiological effects of forest bathing has focused on improved cardiovascular indicators and stress hormones. Multiple studies have shown that forest bathing lowers blood pressure [8,11,12,13,14]. Forest bathing also lowers salivary cortisol, indicating reduced stress levels [2,8,14,15]. Decreased stress leads to positive health outcomes, including improved sleep, immune function, and reduced inflammation [2,8].

Another avenue of research has explored how forest bathing impacts the human immune response. Multiple studies have linked forest bathing to increased levels of anticancer immune cells, sometimes called “Natural Killer” or NK cells [8,14,16,17,18], and decreased inflammatory cytokines [19]. Forest bathing allows participants to breathe in phytoncides, a type of volatile organic compound emitted by trees, which trigger this human immune response [14,16,17,18]. Decreased inflammatory cytokines are linked to decreased symptoms of depression [20,21] and anxiety [22]. Kuo [2] proposed the human immune system as the central pathway that leads to the myriad observed health benefits of spending time in nature and doing activities like forest bathing.

Perhaps this strong link between human well-being and the presence of the central pathway described by Kuo [2] occurs because, when humans spend time in nature, we return to parts of ourselves that are more difficult to access in the modern built world. Some scholars have proposed that this link occurs because humans have an innate and adaptive tendency to respond positively to natural environments, as has been suggested via the biophilia hypothesis [21,23]. According to the biophilia hypothesis, this tendency arises because nature is the environment in which humans evolved [23,24,25]. Although frequently cited as the root of the connection between time in nature and human well-being, Wilson’s hypothesis has been critiqued for being grounded in genetic determinism and leaving out the influences of learning, culture, and individual experiences [26,27,28,29] or the presence of biophobia [28,30,31,32].

Building off the biophilia hypothesis are attention restoration theory [33] and stress reduction theory [34]. Attention restoration theory, which states that natural environments improve mental well-being because they restore people’s ability to pay attention [33], is frequently cited as a framework for the efficacy of nature-based practices like forest bathing [2,26,35,36,37,38,39]. Stress recovery theory, also cited as a reason for the health benefits of time in nature [38,40,41], predicts that if individuals are stressed, time in a natural environment will reduce their stress [34,42]. However, many of the literature reviews summarizing the studies that have supported attention restoration theory or stress recovery theory have provided additional evidence that these theories are likely incomplete explanations of the mechanism through which nature promotes human well-being [2,26,35,43].

Another explanation for nature’s health benefits is that feelings of awe or mystery may mediate positive health outcomes [4,26,44,45]. Much of the research related to nature, awe, and well-being has been related to improving the well-being of people suffering from posttraumatic stress disorder (PTSD). For example, Anderson et al. [44], Poulsen et al. [46], and Westlund [47] pointed to time spent in the outdoors, and in nature in particular, as a potential treatment for PTSD. Participants in these studies, who were adults, reported improved PTSD symptoms after forest bathing [46] or general nature experiences [44,47].

In addition to the innate restorative potential of time in nature, the enhanced efficacy of forest bathing is likely, at least partly, related to a participant’s intent before forest bathing (for example, to relax, receive healing, and connect). This is because the person’s intent significantly affects the outcome of nature visits [48]. Forest bathing practitioners engage in forest bathing with the intent to support their well-being and health.

Although studies have agreed on the statistically significant positive effects of forest bathing on human well-being, data have mainly been collected from research involving young adult males in Asia [8,13,17,18,48]. The few studies involving participants of more than one gender and broader age ranges have shown similar patterns [49,50,51,52]. However, there have been very few studies on adolescents practicing forest bathing. Understanding the potential of forest bathing to positively impact adolescents’ mental well-being is important because, if the health benefits of forest bathing for adolescents are similar to those for adults, then forest bathing is a technique that could be used in schools or other settings to support adolescents’ mental well-being. Therefore, we explored this gap in the research on the effectiveness of forest bathing in improving mental well-being among adolescents in the United States.

### 1.1. Adolescents’ Mental Well-Being

Supporting adolescents’ well-being is necessary because, although adolescence has long been characterized as a time of stressful personal development [53], the rates of anxiety and depression in teenagers increased in the United States by 59% between 2007 and 2017 [54,55], and the rates of hospital admissions for suicidal teens in the United States increased 50% during the same time period [54,56]. These increases are linked to academic and social pressures [54], increased social media and technology use [57], and decreased interpersonal interactions [58,59]. More recently, teenage life during the COVID-19 pandemic was even more stressful than usual [60,61,62,63].

There have been many suggestions for reducing adolescents’ stress and anxiety to improve their mental well-being. These suggestions have included an increased emphasis on social–emotional learning (SEL) in schools [62,64,65]. As the U.S. federal government increased SEL funding in 2020 in the COVID-19-relief CARES Act [66], many schools are adopting or expanding their SEL programs to help their students. Many SEL programs also incorporate contemplative practices like mindfulness and yoga in their efforts to support students [67]. However, few, if any, SEL programs incorporate increasing students’ time in nature to increase their mental well-being despite the well-documented benefits of time in nature in increasing human well-being [3], including the well-being of children and adolescents [6,29,66,68,69].

Although Nisbet et al. [70] conducted a study with young adults to attempt to determine whether practicing walking meditation while spending time in nature had more positive effects than either activity alone, they did not find a significant difference between the three groups. However, the results suggested that combining mindful walking with outdoor time increases mindfulness and attention. Djernis et al.’s [26] review included similar studies that explored practicing mindfulness in nature or alongside nature-based therapeutic interventions. We were curious to see whether a nature-based mindfulness practice, such as forest bathing, had more of an impact on mental well-being than practicing a non-nature-based meditation, such as mindfulness or walking meditation, in nature, as investigated by Nisbet et al. [70] and other authors included in Djernis et al.’s [26] review. In this study, we examined the potential for practicing forest bathing to increase adolescents’ mental well-being, traditionally supported through SEL programs and contemplative practices.

### 1.2. The Practice of Forest Bathing

Forest bathing refers to the practice of spending time in forested areas to invite healing interactions [9]. This nature-based contemplative practice evolved from the Japanese practice of shinrin-yoku. The Japanese Ministry of Agriculture, Forestry, and Fisheries coined the term shinrin-yoku in 1982. It is described as making contact with and taking in the atmosphere of the forest [10], similar to natural aromatherapy [17]. North American forest bathing is a structured walk that invites participants to become embodied and “experience a deeper sense of relationship with the world” [38] (p. 28).

A type of North American forest bathing walk developed by the Association of Nature and Forest Therapy Guides (ANFT) was utilized in this study. It is only one type of forest bathing. This practice is a 1.5–3-h ritualized walk composed of three stages [9,38]. Each stage is designed to create opportunities for embodiment, sensuality, and relationships [9,28]. This sequence of forest bathing, described later, includes many components that support practitioners in experiencing a healing connection with whatever natural environment they choose, although there is abundant evidence that other types of forest bathing practices provide well-being benefits [2,8,11,12,13,14,15,49,50,51,52,71].

### 1.3. Current Study

The purpose of this study was to determine the impact of practicing forest bathing on adolescents’ mental well-being. Additionally, we wanted to explore adolescents’ experiences of forest bathing, as most of the forest bathing research has studied quantitative measures of mental and physical well-being. The participants in this study, who were high school students in the first author’s advanced placement (AP) environmental science class, partially co-created the convergent parallel mixed-methods study as part of a youth participatory action research (YPAR) project on improving mental well-being to answer the following research questions:Does practicing forest bathing impact adolescents’ mental well-being?What are adolescents’ experiences of practicing forest bathing?

During this study, we measured mental well-being before and after practicing forest bathing one time and three times with the Warwick–Edinburgh Mental Well Being Scale (WEMWBS). We also documented participants’ lived experiences through journaling, responses to open-ended prompts that we added to the WEMWBS, and photovoices after forest bathing.

## 2. Materials and Methods

### 2.1. Participants and Sites

This research took place in the spring of 2022 at Southampton High School, located on the ancestral lands of the Shinnecock Nation in what is now Southampton, New York, USA. Southampton High School is the high school for a small suburban school district in which over 50% of students are eligible for free or reduced lunches because of low family incomes [72]. The participants were twenty-four 11th- and 12th-grade students in the first author’s AP environmental science classes. Each year after the AP exam in May, students complete a student- or class-chosen action research project to conclude their year of study. In the spring of 2022, students completed an in-class YPAR project on students’ mental well-being, particularly concerning how spending time in nature might alleviate students’ stress and anxiety. We chose to focus on mental well-being since Spring 2022 was the second year of the COVID-19 pandemic, and students reported increased stress and anxiety levels. Students were intrigued by a 2021 pilot study investigating the mental health benefits of forest bathing that previous AP Environmental Scienceenvironmental science students had completed, and the current students wanted to continue the work.

All 24 participants completed the first 2 weeks of the forest bathing series and forest-bathed twice. Sixteen participants completed all 3 weeks of the series and practiced forest bathing a total of three times. This research underwent an Institutional Review Board review and was approved by New England’s Institutional Review Board committee at Antioch University. The participants or their parents/guardians could choose to have their data excluded from this study with no academic or other penalties.

This study took place in the first author’s classroom and three nearby nature preserves (see Figure 1). One nature preserve is in an area called Wickapogue, which in Algonquian means “end of the water” or “end of the land” [73]. In 1986, Wickapogue was renamed the Richard Fowler Nature Preserve by Southampton Village [74] to honor Richard Fowler, a former Southampton Village trustee. We chose this site because it is very close to Southampton High School and would maximize forest bathing time. This location also provided a mix of closed- and open-canopy forests and a small pond shoreline, which offered a variety of spaces for partnership invitations.

Another nature preserve where we practiced forest bathing was the Wolf Swamp Preserve, near a lake now called Big Fresh Pond. This lake is called Missapogue in Algonquian, which means “large lake” or “large water” [73,76]. We could not determine when the lake’s name officially changed. The New York State Department of Environmental Conservation and a local homeowner’s association still use Missapogue to refer to the lake [76,77,78]. The Wolf Swamp Preserve is on the northwest side of Big Fresh Pond. Elizabeth Morton Tilton donated this 20-acre preserve to the Nature Conservancy in 1957 [74]. Despite its name, there are no wolves in Wolf Swamp Preserve, as wolves were hunted to extinction on Long Island due to a bounty system established by settlers in the 1600s and 1700s [79,80]. Like the Richard Fowler Preserve, this site also provides a mix of closed- and open-canopy forests, as well as the lake’s shoreline. There were loud home construction noises at Richard Fowler on our first visit. Hence, the participants and the first author decided that the study’s second time practicing forest bathing should take place in an area away from development. We chose Wolf Swamp Preserve since it is more isolated than the Richard Fowler Preserve yet close enough to Southampton High School that there would be enough time for forest bathing.

The third location Is part of Southampton Village’s Old Town Pond Park and Beach. The area is called Old Town because it was the location of the first English settlement in Southampton. It is an ocean beach with a grassy and shrubby open foredune environment. Although this is not a forested location, this location is suitable since forest bathing can be practiced in any environment [8,51,52,81,82]. Although our original intent was to return to the forest in the Wolf Swamp Preserve, several participants found ticks on themselves after forest bathing in Wolf Swamp Preserve. The first author decided that choosing a location with a low probability of tick activity was essential for one final time practicing forest bathing before school ended for the summer. This was to minimize the potential for tick-borne diseases and decrease potential participant drop-out because of the fear of ticks.

### 2.2. Mixed Methods

Creswell and Plano Clark’s [83] mixed-methods participatory social justice design was used to investigate the following research question: How does practicing forest bathing impact adolescents’ mental well-being? This design adds mixed-methods research to a participatory or social justice perspective to involve participants in the research and foster changes in individuals, institutions, or communities [83]. Mixed-methods participatory social justice, sometimes called transformative mixed methods [84,85,86,87], is useful for identifying and describing issues and including the voices of participants while also generating evidence that is useful and persuasive to individuals and stakeholders [83,88]. The collaborative intent of participatory research provided a framework for conducting aspects of mixed-methods research in the first author’s classroom. In the case of this study, a convergent, parallel, mixed-methods design was embedded into a YPAR methodological framework. YPAR is a democratic approach to research in which participants work collaboratively with a researcher in the creation of new knowledge to address a specific issue or problem [89]. YPAR provides a space for young people to become researchers on their own experiences [90]. In addition, using YPAR allowed us to gain a more authentic and nuanced insight into the research questions than we could have if the research had been conducted through a more hierarchical and traditional approach. The YPAR project was designed and conducted with the first author’s students to investigate their experiences of practicing forest bathing and to determine whether forest bathing impacted their mental well-being.

A convergent parallel design was chosen because the design allowed us to obtain “different but complementary data on the same topic” [91] (p. 122), in this case, the impact of forest bathing on adolescents’ mental well-being. In a convergent design, the qualitative and quantitative data collection and analysis are considered separate [88]. This is based on the assumption that qualitative and quantitative data provide different types of information. The qualitative data revealed the depth of participants’ experiences, and the quantitative data revealed the breadth of these experiences.

### 2.3. Quantitative Methods and Instruments

The WEMWBS was used to determine participants’ mental well-being. This tool is composed only of positively worded items relating to different aspects of mental health [92]. It was developed in part to support the evaluation of mental well-being programs [92]. The WEMWBS is responsive to relatively small samples [93], which was necessary for this study with 24 participants.

The WEMWBS has been extensively validated to be responsive to those aged 13 and older in multiple cultures [92,93,94,95,96,97], and it has been translated into 28 languages [96,97]. It was important to connect this study with other research, whether other forest bathing [98] or nature exposure research [99] that used the WEMWBS or other research on interventions to improve adolescents’ mental well-being.

The WEMWBS is quick and easy to complete, with only 14 items scored on a 1–5-point Likert scale. This was essential, as the time for forest bathing and post-evaluation was limited to 90 min by the school’s schedule. The WEMWBS has been shown to provide a credible picture of mental well-being, it is sensitive to changes that can occur in mental well-being in populations, small groups, or individuals [93,97], and it is considered suitable for measuring change due to interventions or programs [96,100].

The WEMWBS is scored by summing the response scores for each item. The minimum score is 14, indicating lower levels of mental well-being, and the maximum is 70, indicating higher levels of mental well-being. The range in WEBMWBS score changes that is considered meaningful varies from 3 to 8 [93,100]. However, Maheswaran et al.’s [93] review found that a change of 3 points or more in an individual’s WEMWBS score could be interpreted as important. The WEMWBS has also been shown to be helpful as a screening tool for depression, with a cutoff total mental well-being scores of <42 indicating probable clinical depression and total mental well-being scores of 41–45 indicating possible or mild depression [92,93,96].

### 2.4. Qualitative Methods

The participants determined the study’s qualitative methods during their in-class YPAR project. These methods included open-ended survey responses, journal entries, and photovoices. Open-ended survey responses and journaling helped the participants document and reflect on their work as researchers and their experiences with forest bathing. In addition, during the research design, the participants and the first author discussed participant response and researcher bias. Although essential for any research project, this discussion was particularly relevant in this study since the participants were the first author’s students, which constitutes a limitation of this study, and co-researchers. The participants were told they should be truthful and authentic in their responses about their experiences and that the data would be de-identified.

Participants wanted to add a place for free response writing at the end of the WEMWBS to gather information that the survey might have missed about participants’ experiences. Participants also wanted to provide a more extended journaling opportunity. Journaling is a reflective practice through which practitioners can describe and reflect on their feelings and lived experiences of an event or occurrence [101,102]. Journaling is often used in experiential learning cycles [102,103] and in action research [104]. We also chose journaling because it is a time-efficient and effective method of gathering data on participants’ personal experiences [105,106,107]. Although journaling can be free-form, we created prompts for the open-ended survey question and journal entries to help guide participants’ reflections on their experiences practicing forest bathing.

The participants collaboratively created the open-ended survey response and journal prompts with each other and the first author. At the end of the pre-forest bathing survey, the participants responded to the prompt “Describe what you imagine your forest bathing experience will be like”. At the end of the post-forest bathing surveys, the participants responded to the prompt “Describe what your forest bathing experience was like, including any impacts on your well-being or connection to nature you may have experienced”. The prompt for post-forest-bathing journaling was “Describe your experience of forest bathing and compare your experience to what you thought the experience would be like”.

The final type of qualitative data that participants generated were photovoices. Photovoice is a qualitative method in which participants use photographs to document their experiences [108,109,110]. Participants can use their photos to identify, highlight, and represent issues of importance to them. They can write about or discuss their photographs, which can be shared with others if desired. Photovoice enables researchers to understand the issue under study [111,112,113]. The participants used their phones to take photos in Wolf Swamp Preserve after the second time practicing forest bathing. Their prompt was “What would you want to share about your forest bathing experience?” They were also invited to complete another photovoice after practicing forest bathing for the third and final time at Old Town Park Beach. Back in the classroom, the participants selected one photo that best showed their forest bathing experience. They wrote captions about their photographs, describing their experiences of forest bathing, and they responded to the following prompt: “What are you showing the viewer about the forest bathing experience?” Seeing participants’ photos and reading their captions gave us a more authentic and nuanced insight into their experiences than we would have gained from only the survey results and observations while practicing forest bathing.

### 2.5. Procedure

Figure 2 presents the flow of the study’s data collection and procedure. Data were collected before forest bathing, after forest bathing once, and at the end of the entire 3-week forest bathing series, once forest bathing had been practiced the third time. Data were gathered after one forest bathing experience because some members of the forest therapy guide community were curious about the impact of a single forest bathing walk [40]. The first author wanted to determine how changes in mental well-being after one walk compared to any changes that occurred after completing a series of walks over 3 weeks.

To determine their mental well-being before forest bathing, the participants completed the WEMWBS survey with a free-response prompt in Google Forms during the class the day before. Although this constitutes a limitation of this study, the participants completed the survey the day before because of the school district’s time restrictions on field trips to practice forest bathing. The next day, the first author, who is an ANFT-certified guide, guided the participating students through a 90-min shortened version of the ANFT forest therapy sequence for forest bathing, adapted from Clifford [9] and Page [38], at Wickapogue.

The first walk began inside the Richard Fowler Preserve and finished with a sit spot and tea ceremony at Wickapogue Pond. After completing forest bathing, all 24 participants immediately completed post-WEMWBS surveys in Google Forms on their phones. They also responded to the prompt that had been added to the end of the WEMWB survey and journaled about their experiences. Participants who could not access the Google Form were given a paper copy of the survey to complete. Completing the survey, responding to the prompt, and journaling took the participants approximately 10 min. Before completing the WEMWBS and questions, the first author reminded the participants that the data would be de-identified and that they should respond truthfully about their experiences.

Six days later, the 24 participants and the first author went to the Wolf Swamp Preserve to practice forest bathing for the second time, following the same sequence as the first time. After completing the final tea ceremony, the participants were afforded 10 additional minutes in the nature preserve to use their phones as cameras to document their most meaningful forest bathing experiences through photovoice. Before taking pictures for their photovoices, the participants were reminded that their names would be removed from the photovoice and that they should document whatever was most meaningful to them. The participants chose to take pictures of themselves, the nature preserve, and each other in the nature preserve. The next day, the participants each selected one photo, gave voice to the photo by writing a caption describing the selected photo, and then uploaded the photo and caption to a Google Doc. The participants also added their photovoice to a shared Google Slides presentation to be shared with the whole class, the school, or on social media.

The third forest bathing experience took place 5 days after the second on the ocean beach of Old Town Park. The first author again led the participants who could attend (16 this time) in practicing forest bathing by following the same sequence as the first and second times. The study participants who did not attend this outing were absent from school. After completing the final forest bathing tea ceremony, the participants immediately completed post-WEMWBS surveys and the prompt we had added to the end of the WEMWBS survey. Participants who could not access the Google Form on their phones were given a paper copy of the survey to complete. After completing the surveys, the participants were also invited to complete another photovoice using their phones as cameras and to journal about their forest bathing experiences.

## 3. Results

### 3.1. Quantitative Analysis

To determine significance, the Microsoft Excel [114] descriptive statistics function was used, confirming that the data were normally distributed, with low skewness and kurtosis, so that the appropriate types of analyses could be selected. As the data were normally distributed, a one-factor ANOVA was performed to determine whether there were significant differences among all three groups. A Tukey–Kramer post hoc test was used to examine pairwise differences between groups because the number of participants fell from 24 to 18 between the second and third times forest bathing, so there were different sample sizes. An alpha level of 0.05 was used to indicate statistical significance.

Hedges’ *g* was used to determine the effect size of practicing forest bathing. Hedges’ *g* is considered appropriate when evaluating responsiveness in single-group pre–post studies, particularly when comparing samples of different sizes [115] or small sample sizes [116]. Hedges’ *d* was also chosen since it is an effect size statistic commonly used to determine the effect of medical and psychological interventions [115,117,118], and we wanted the results of this study to be comparable with other studies on adolescents’ mental well-being.

### 3.2. Quantitative Results

As measured using the WEMWBS, the mean total mental well-being significantly increased after the participants had practiced forest bathing (see Figure 3). A single-factor ANOVA of all three groups showed that the difference among all three was significant (*p* < 0.01). After one time practicing forest bathing, the mean mental well-being increased from the baseline pre-test score of 47.8 ± 3.2 to 54.3 ± 4.0 (*p* < 0.01). This 6.5-point increase shows a moderate effect size (Hedges’ *g* = 0.7) and is considered a meaningful improvement using the WEMWBS [93,100]. Practicing forest bathing three times increased the mean mental well-being to 56.3 ± 4.0. This indicates a large effect size (Hedges’ *g* = 1.0) and is significant compared to the baseline pretest score (*p* < 0.01). Although there was no significant difference in the mean mental well-being between one time and three times forest bathing (*p* > 0.5), the small increase in mental well-being between the first and third times forest bathing is still considered a meaningful improvement using the WEMWBS [93,100].

In addition, the pre-forest-bathing mental well-being mean of 47.8 was above the WEMWBS thresholds for possible/mild depression (41–45) or probable clinical depression (<41). However, based on the pre-forest-bathing survey, three participants met the threshold for possible/mild depression, and six met the threshold for probable depression on the WEMWBS. After practicing forest bathing once, no participants met the threshold for possible/mild depression, and the number of participants who met the threshold for probable depression decreased to three. Because eight participants did not complete the third forest bathing practice, we were not able to determine whether practicing forest bathing three times would further decrease the number of participants who met the threshold for probable depression.

### 3.3. Qualitative Analysis

To analyze the qualitative data, the participants’ journal entries and their responses to the end of the survey prompts were printed. The first author then read through the texts. The responses to the end of the pre-survey prompt were used to analyze the participants’ expectations of their forest bathing experiences. To analyze the participants’ descriptions of their forest bathing experiences, the participants’ post-one-time-forest-bathing and post-three-times-forest-bathing journal entries and responses to the end of the survey prompt were used. The second time reading through the text, the first author and her YPAR co-researchers separately highlighted specific words and phrases, made notes, and formed initial codes. The first author also coded the participants’ captions and descriptions while viewing each photovoice in Google Docs to analyze the photovoice data, and some YPAR co-researchers also chose to do this.

After the YPAR concluded at the end of the school year, the first author created three documents. All the participants’ anonymized responses to the end of the presurvey prompt were placed in one document. Then, the process was repeated for all the post-one-time-forest-bathing and post-three-times-forest-bathing journal entries and participants’ responses to the end of the post-survey prompt. All the documents were separately uploaded into NVivo [119] software, and a second coding round was conducted in nVivo. During coding on paper and in NVivo [119], the first author looked for similar patterns and themes. Three themes emerged: relaxation and peace, mindful qualities, and being away. Being away is a phrase derived from Kaplan’s work on attention restoration [33]. These are described below, with all participants’ names changed to pseudonyms.

### 3.4. Qualitative Findings

#### 3.4.1. Relaxation and Peace

The most common coding of the participants’ journals and open-ended question responses related to relaxation and peaceful feelings. After one time practicing forest bathing, 22% of the text was coded to the theme “relaxation and peace”, which increased to 30% after forest bathing three times. Many participants, like José, explicitly described their experiences practicing forest bathing as relaxing (“My experience of forest bathing was relaxing”) or, like Annie, as calming (“I felt great about the experience. It really calmed me down”). Relaxation can be related to stress reduction, as Luz wrote: “It benefited my peace of mind and any anxieties and stresses I had were gone”.

Some participants, like June, linked their feelings of relaxation to the sit spot invitation (Figure 4). The feeling of relaxation that some participants experienced was also linked to a desire to continue the practice. For example, Gloria wrote, “Forest bathing is definitely something that I plan on continuing because it was just so relaxing”.

In addition to relaxation, many participants described their forest bathing experience as helping them feel peaceful. For example, Alexandra wrote that she “felt way more calm and peaceful”, and John wrote that he “felt as though l was completely at peace”. Some participants, like Sara, linked their feelings of peace to a connection to nature, as evidenced when she wrote, “I felt a greater connection to not only nature but all the peace within myself”. For others, this peaceful feeling was linked to experiencing forest bathing as spiritual. Olive expressed this when she wrote, “Forest bathing was very positive in calming my mind and soul”. Likewise, John wrote, “I didn’t feel stressed nor quick to temper. It’s like I had changed inside. It was relaxing, fun, and even spiritual”. All these post-forest-bathing descriptions reflect how participants responded to the open-ended prompt in the presurvey. For example, pre-forest-bathing, Jay and Luna wrote, “I think my forest bathing experience will be very relaxing” and “I think I will feel happier and calmer”, indicating their anticipation of feeling relaxed and calm after forest bathing.

#### 3.4.2. Mindful Qualities

Mindful qualities, which, for this study, included expressions of gratitude, present moment awareness, and slowing down, increased from 14% after practicing forest bathing one time to 23% after the third time practicing forest bathing. Participants described slowing down and being more aware in both their written responses and the photovoice captions of their photos, often including specific invitations when describing their experiences. For example, Kathy described her photovoice (Figure 5) as follows:
What’s in Motion… helps us slow down and change our pace so you can notice more of what’s right in front of you. This is beneficial because depending on what you see or what you find it can help you calm down and connect with nature.
In Kathy’s case, slowing down was also linked to calm and the ability to connect with nature. This was similar to Bri’s description:
I was able to fully connect with myself with nature and become very aware of the surroundings. I was able to notice things such as the ways the plants swayed in the wind and the different creatures living within the forest.

June explained that she “sat on a nice bridge as well as picked a tree to get to know”. Likewise, Sara also wrote about being more aware of her surroundings while slowing down: “I’ve surprised myself with how much I’ve overlooked, noticing all the details that have always been there have helped me slow down and enjoy moments like these”. Ernesto also connected slowing down with feeling calmer: “If you just take your time and slow down your life for a moment you feel a sense of relief”.

The participants’ descriptions related to mindful qualities also included perceptions of oneness with nature and awe. Mel wrote, “I felt extremely relaxed and truly one with nature. I felt as though l was completely at peace with my existence and one with the world”. Jace wrote, “I realized how much I truly care about the things in nature and now feel like I am connected to the web of life”. Participants also wrote about their gratitude and concern for nature; for instance, Brian wrote, “After three experiences of forest bathing, I feel way more appreciative of nature”. And Kelly wrote, “All forms of nature are important in their own ways. It is important to know that all ecosystems deserve to be protected”. More gratitude, this time for practicing forest bathing, was expressed by Valeria: “I am very thankful to have gotten the chance to further connect with nature”.

#### 3.4.3. Being Away

Descriptions and photovoices reflecting the being away theme included statements describing forest bathing as a break from school or life, developing a clearer mind or thinking while forest bathing, and feelings of enjoyment, fun, or play. The percentage of participant responses related to being away stayed steady at 21% even after practicing forest bathing three times. Being away was evident in Larry’s photovoice (Figure 6), both in his photo and the description, where he wrote about forest bathing “being able to be somewhere that has not been rearranged by society”. Some participants described how their experiences practicing forest bathing felt like a break from their stressful lives; for instance, Jessie wrote that they were “feeling tired and emotionally done with school. Afterward, I felt a sense of relief and relaxation that remained with me for the rest of the day”. Several participants expressed similar feelings; for example, Laurey wrote that “school can be very stressful, and forest bathing provides an escape into nature where one can relax”, and Carla “was happier, less tired, and less stressed afterward. It was a nice break in my day to relax”. There were many overlaps between the being away theme and the relaxation and peace theme.

This feeling of being away and having a break helped some participants clear their minds; as Mariana and Julie, respectively, wrote, “I was able to clear my mind and think and focus”, and “I talked to a rock which sounds silly but it helped organize my thoughts”. And some participants, like Eduardo, mentioned disconnecting from technology as an important part of their experience: “It was very relaxing to disconnect from the online world and clear my mind”. Others, such as Luke, wrote about how forest bathing helped them get back to themselves: “I feel like my energetic frequency is higher, my mind is more awake”. And Trinity wrote, “I honestly think being in nature reflects my true emotions”.

For several participants, like Jay and Linda, the feeling of a break was described as enjoyment: “I enjoyed the nature around me and felt that spending time outside improved my mood”, and “I enjoyed being able to not have to worry about anything”. This enjoyment, for some participants, like Amalia, was linked to nature awareness: “I enjoyed listening and feeling the nature around me”. And Kaily wrote, “It was fun connecting with nature. I liked meeting a tree”. For others, like Andy, it was linked to practicing forest bathing with others: “Also, it was nice to have friends around you as well to make the experience more happy”.

## 4. Discussion

The goal of this study was to understand the impact of practicing forest bathing on adolescents’ mental well-being and to explore adolescents’ experiences of forest bathing. The 6.5-point increase in the mean mental well-being from the baseline pre-test score of 47.8 ± 3.2 to 54.3 ± 4.0, as measured using the WEMWBS survey, suggests that practicing forest bathing, even just once, significantly increases adolescents’ mental well-being (*p* < 0.01) with a moderate effect size (Hedges’ *g* = 0.7). Practicing forest bathing three times had a large effect size (Hedges’ *g* = 1.0) and further increased the participants’ mean mental well-being by two points to 56.3 ± 4.0. These results align with previous forest bathing research involving adult [2,8,11,12,13,14,15] and adolescent [71] participants.

In addition, the participants’ descriptions of their experiences practicing forest bathing indicated that they found these experiences relaxing, that forest bathing increased their mindful awareness, and that they felt like they had gotten away from their everyday experiences. All three of these traits have been linked in previous research to improved mental well-being [120,121,122,123]. Three primary factors could contribute to the mental health benefits of forest bathing. Factors include the structure of forest bathing walks, setting an intention for forest bathing walks, and forest bathing taking place outside, in nature.

### 4.1. The Components and Structure of the Forest Bathing Walk Support Improved Mental Well-Being

First, the structure of this type of forest bathing walk creates more opportunities for open awareness than merely spending time in nature. This is important because open awareness is a state shown to improve mental well-being [124,125]. Forest bathing creates these opportunities through a sequence that contains many elements known to improve mental well-being.

As mentioned in the introduction, this type of forest bathing walk is structured in three stages [38]. The first stage uses sensory connection to shift participants’ awareness from their thinking minds to their senses, the present moment, and the present place. This is accomplished in three ways. First, participants check in with their bodies, which is shown to increase mental well-being [126,127]. Next, participants are guided through their senses and invited to notice what they are experiencing to encourage open awareness, another technique that has been shown to increase mental well-being [124,125]. In the final part of this first phase, called “what’s in motion”, participants are invited to slow down and notice what is moving in the forest. Again, slowing down and increasing open awareness increases mental well-being [128,129]. Participants journaled how slowing down made them more aware of the nature preserve, with Kathy describing how forest bathing “helps us slow down and change our pace so you can notice more of what’s right in front of you”, whereas Sara wrote how she was “noticing all the details that have always been there have helped me slow down and enjoy moments like these”.

Other vital elements of forest bathing support participants’ improved mental well-being. For example, in the second part of the forest bathing walk, the guide offers participants several open-ended partnership invitations [9,38,40]. These invitations support improved mental well-being, perhaps by generating happy feelings [130,131,132] through the curiosity, play, and creativity that can arise when participants explore these invitations. Partnership invitations are open-ended and allow participants autonomy to interpret their experiences in their own ways. Feeling autonomous is also linked to mental well-being [133,134,135]. Descriptions of the various partnership invitations filled many participants’ journal entries, such as “I talked to a rock which sounds silly but it helped organize my thoughts”, and “I sat on a nice bridge as well as picked a tree to get to know”.

After each invitation, the participants form a circle, and everyone has a chance to respond to the prompt “What are you noticing?” The participants share with the whole circle while the rest of the group listens, or they share in a pair-share format. Everyone has a turn to share their stories and have their stories witnessed by others multiple times in a walk. The sharing circle allows each participant to share and listen, observing the simultaneous commonality and uniqueness of everyone’s experiences. The power of having our stories heard and hearing others’ experiences [136,137,138] is also an aspect of forest bathing that supports improved mental well-being. In addition, the sharing circle also provides opportunities for positive social interactions, which have also been shown to support adolescents’ mental well-being [58,139,140,141].

The final partnership invitation is a sit spot, where participants find a place alone to meditate quietly. Meditation is a mindfulness practice that has long been shown to improve the mental well-being of its practitioners [127,142,143]. Meditation in sit spots provides opportunities for participants to begin to integrate their forest bathing experiences. Integrating transformative experiences, whether using psychedelic medications or rites of passage, improves the effectiveness of these experiences [144,145,146,147,148]. Photovoice shows that the participants valued the sit spot part of forest bathing. They not only chose to capture the moment with a picture but also chose to write, as their caption, for instance, “I am showing the viewer where my sit spot was and how relaxed I was while sitting. During this picture, I was enjoying the sunlight and birds chirping”.

Integration is further facilitated in the final part of a forest bathing walk. This final part is marked by a tea ceremony ritual prepared by the guide. Here, participants gather one last time to share tea, food, and stories and to express gratitude. Expressing gratitude, such as during the tea ceremony, has been shown to increase happiness and motivate self-improvement and positive change [149,150,151,152], including in adolescents [144,153]. Many participants in this study expressed gratitude for the nature preserves in which they practiced. For example, “This experience gave me a deeper appreciation for nature”, and “I am very thankful to have gotten the chance to further connect with nature”. In addition, participants like Kay also wrote about gratitude for the forest bathing practice: “The forest has a very calming atmosphere that slows me down from the rapid pace of everyday life, and I’m grateful that I was able to witness that firsthand through these trips!” Kay’s words also show how practicing forest bathing creates a feeling of being away from one’s everyday experience, which has been shown to be restorative [29,33,35,43,66,148,149,150,151,152,153,154,155,156].

### 4.2. Setting an Intention for the Walk Supports Improved Mental Well-Being

Second, in contrast to going for a nature walk or spending time in nature, practicing forest bathing involves an intention to relax and cultivate a healing connection with the forest or nonhuman beings wherever it is practiced. Setting this intention, which took place in this study during the YPAR project with the participants, may be another factor that facilitates increased mental well-being after forest bathing. This is because setting an intention before an experience can influence the outcome [157] or perceptions of the outcome for both adults [158,159] and adolescents [160].

More specifically, Pasanen et al. [41] found that a person’s intent significantly affects the outcome of nature visits. In other words, because forest bathing practitioners engage in forest bathing with the intent to support their well-being and health, it does support their well-being and health. In this study, participants showed this intent when they responded to an open-ended prompt in the presurvey. For example, Jay and Luna wrote, “I think my forest bathing experience will be very relaxing” and “I think I will feel happier and calmer”, indicating their anticipation of feeling relaxed and calm. Anticipating these positive mental well-being effects could be at least partially responsible for the significant increase in the measured adolescent mean mental well-being after forest bathing. This aligns with other research involving adolescents that has also shown that, when adolescents set an intention before an activity, it influences both the outcome and perceptions of the activity [160].

### 4.3. Forest Bathing Takes Place Outside in Nature, Which Supports Mental Well-Being

Although it seems obvious, a third explanation for the efficacy of forest bathing is that it takes place outside, often in natural environments. However, in addition to forests, forest bathing can be practiced in any environment, including more built environments like urban parks or hospital and nursing home gardens [8,51,52,143,144]. In this study, the participants were guided through the forest bathing sequence twice in a forested environment and once in an open beach environment (see Figure 1). In other words, throughout the 3-week forest bathing series, the participants were exposed to green and blue spaces, both of which have been shown to improve mental well-being [4,5,6,7]. Spending time in these green and blue spaces may have been beneficial because of the innate properties of these natural environments [1,2,3,4,5,6,7,145] or because the participants were getting away from their school environment, which has been theorized to restore attention [24,27,29,68,69,161,162,163].

Numerous studies have shown that spending time in nature increases human well-being [2], particularly mental well-being, including alleviating the symptoms of anxiety and depression [2,3,83] and posttraumatic stress disorder [37,146]. In their review, Bratman et al. [83] used that research to support their proposal that mental health be included as an ecosystem service provided by nature. Shanahan et al. [16] recommended nature-based interventions, including increasing access to green spaces, to improve community and organization mental well-being. This recommendation was particularly relevant to this research, which occurred during the COVID-19 pandemic, when levels of adolescent anxiety, depression, and posttraumatic stress disorder increased [60,61,62,63,162]. Data from the WEMWBS gathered in this study not only showed significant increases in mean mental well-being over the forest bathing series but also indicated that the number of participants meeting the WEMWBS threshold for depression decreased after practicing forest bathing once. One participant journaled about their forest bathing experiences as follows: “They were relaxing and calming. It benefited my peace of mind, and any anxieties and stresses I had were gone”.

In addition to suffering from general anxiety and depression, many adolescents suffer from a particular type of anxiety caused by environmental degradation called eco-grief or eco-anxiety [164]. Eco-anxiety symptoms include a low mood, disturbed sleep, panic attacks, and feelings of anger, guilt, or helplessness [164]. Spending time in nature can be a way for adolescents to relieve their eco-grief or eco-anxiety by experiencing more intact ecosystems and perhaps learning to care for their local places [6,68,69,150,151,152,153,165,166]. Participant responses included expressions of care and concern for nature: “I realized how much I truly care about the things in nature and now feel like I am connected to the web of life”, and “All forms of nature are important in their own ways. It is important to know that all ecosystems deserve to be protected”.

Furthermore, spending time in nature is also a potential conduit for mindfulness and experiences of awe, both shown to improve mental well-being [44,121,167]. Lawlor [121] stated that “Moment to moment awareness when in nature, focusing attention on sights, sounds, and smells, may encourage inner stillness, contemplation, and gratitude” (p. 70). Deringer [167] explored mindfulness through the lens of outdoor adventure education. They suggested that outdoor education should emphasize a connection to place through mindfulness instead of only emphasizing experiences. This is particularly relevant for this study with adolescents, as much research has supported a connection between adolescents’ mindfulness and their mental well-being [67,121,135]. Many participants in this study, such as Bri, expressed increased mindfulness and connection:

I was able to fully connect with myself with nature and become very aware of the surroundings. I was able to notice things such as the ways the plants swayed in the wind and the different creatures living within the forest.

### 4.4. Limitations and Suggestions for Further Research

Although this investigation used a small sample size and the participants practiced an abbreviated forest bathing walk during each of the three sessions over 3 weeks, these findings suggest that practicing forest bathing can improve adolescents’ mental well-being and that adolescents practicing forest bathing is an important area of research that merits further exploration. Exploring the difference between adolescents practicing forest bathing in nature and those spending time in nature in a controlled experimental study is a logical next step to more specifically identify the practice’s effects. It would also be helpful to have participants complete pre-surveys immediately before forest bathing to obtain more accurate pre–post-WEMWBS scores. In addition, it would be interesting to study the impact of practicing forest bathing for a longer time, such as practicing for more than 90 min or for more than 3 weeks. It is also important to note that, over 3 weeks, many things can change in an adolescent’s life that can cause changes in mental well-being, not just practicing forest bathing. Also, to further strengthen our understanding of forest bathing’s impact on adolescents’ mental well-being, future research could include participants who are not a researcher’s students, which could minimize participant bias, as well as adolescents in additional settings. It would also be useful to determine how long the mental health benefits of forest bathing last to recommend how frequently forest bathing must be practiced in order to maintain elevated mental well-being.

## 5. Conclusions

Anne Frank believed that “nature brings solace in all troubles”. As schools and communities are searching for ways to alleviate the epidemic of adolescents’ mental suffering from anxiety and depression, they are adapting new SEL standards and programs. Practicing forest bathing could become part of an SEL toolkit. Forest bathing does not require special equipment, and it increases adolescents’ mental well-being and feelings of mindfulness, relaxation, and peace. In other words, nature brings solace, something many adolescents desperately need.

## Figures and Tables

**Figure 1 ijerph-21-00008-f001:**
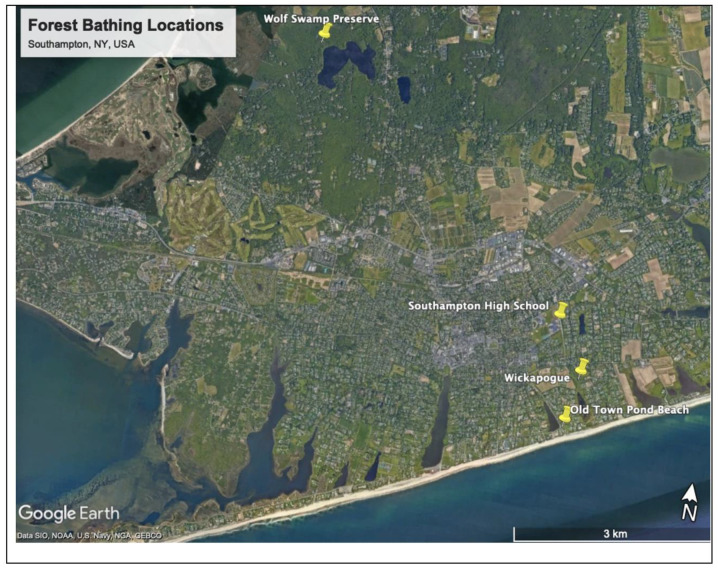
Forest Bathing Locations in Southampton, NY. Source: Google [75].

**Figure 2 ijerph-21-00008-f002:**
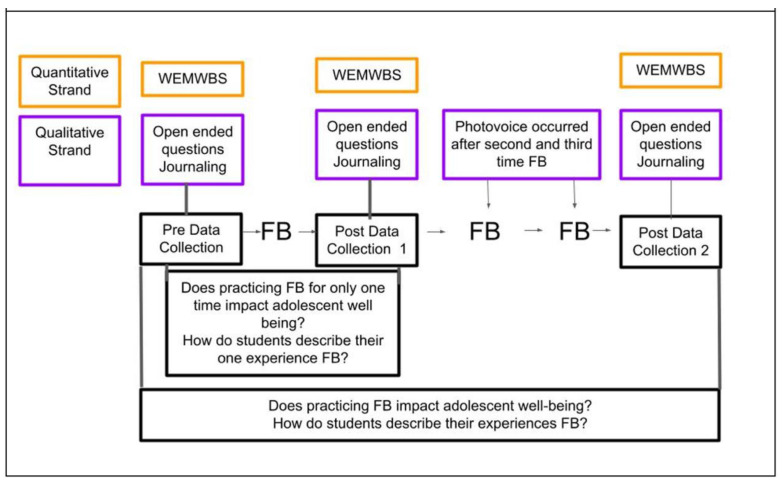
Procedure for convergent parallel mixed methods.

**Figure 3 ijerph-21-00008-f003:**
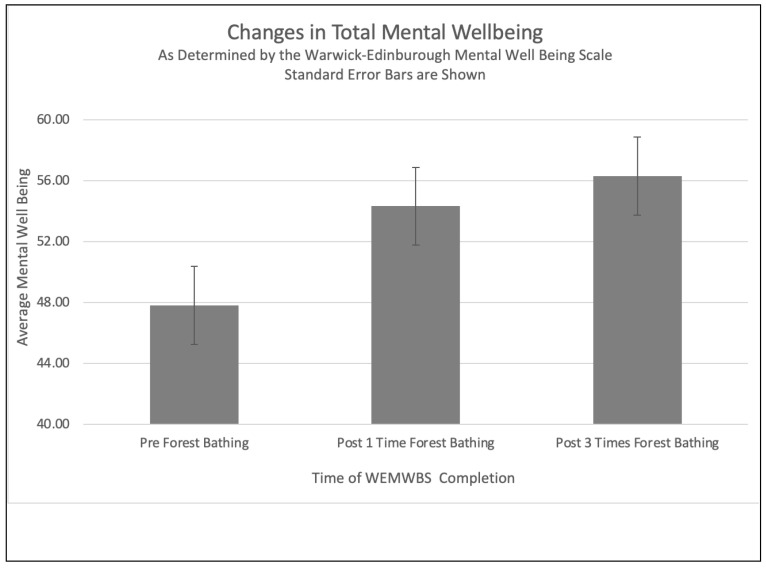
Graph depicting changes in mental well-being.

**Figure 4 ijerph-21-00008-f004:**
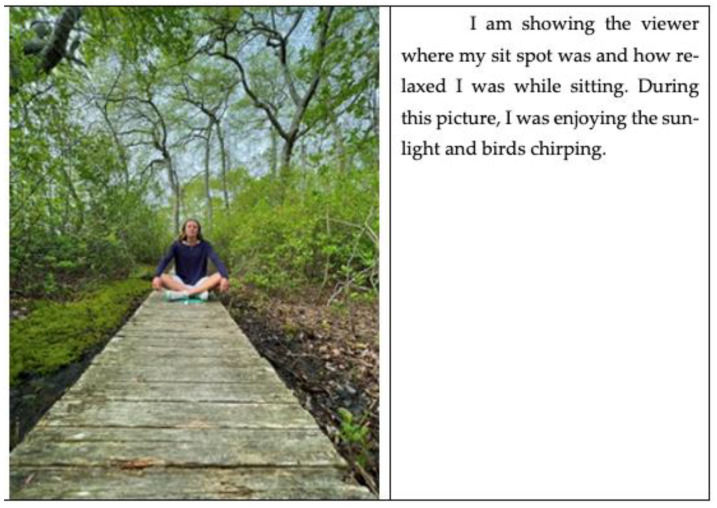
June’s photovoice.

**Figure 5 ijerph-21-00008-f005:**
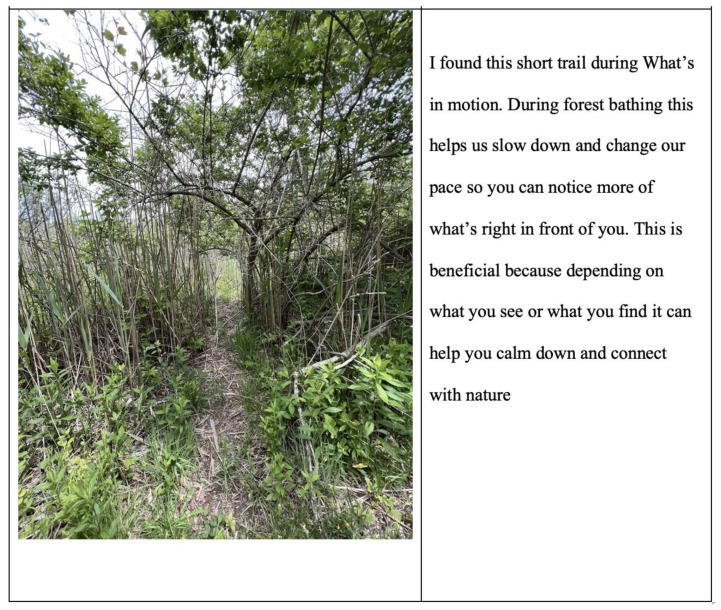
Kathy’s Photovoice.

**Figure 6 ijerph-21-00008-f006:**
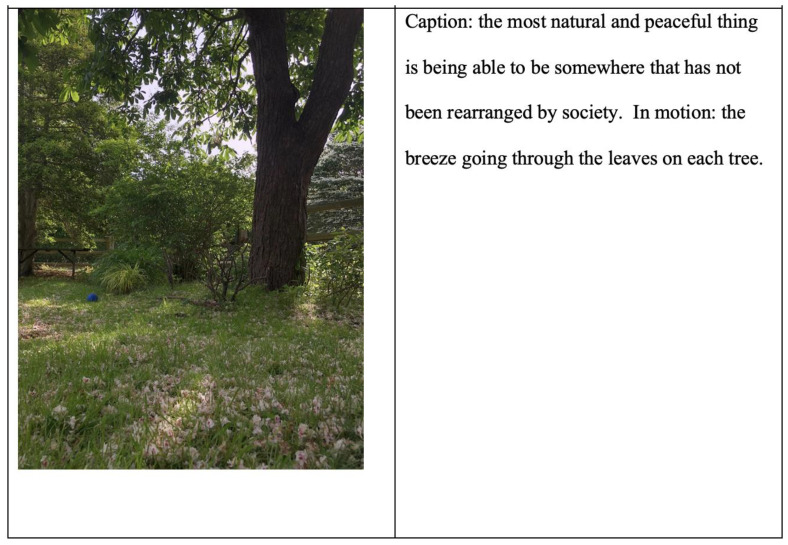
Larry’s photovoice.

## Data Availability

The data presented in this study are available on request from the corresponding author.

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
