# Peer review of "Forest Bathing Increases Adolescents’ Mental Well-Being: A Mixed-Methods Study"

_ijerph, 2023, doi:10.3390/ijerph21010008_

Round 1
Reviewer 1 Report
Comments and Suggestions for Authors
This is a mostly well-designed study that accurately identifies a gap in the literature regarding the target population (adolescents) and the need to address well-being in this population. Good attention is paid to procedural and potential ethical issues, though there are places were more attention to the introduction of potential bias would be helpful. Employment of mixed-methods is an underutilized design in this area of research, and was appropriately employed in this study. The authors make claims of the findings of the study which are not supported particularly due to lack of a control group which is necessary to establish causality.
1. Inclusion of the immune system/Kuo 2015 material as written is tangential. There are no measures of immune system function in this study. Either describe how mention of NK cell/phytoncide function is relevant for the well-being states being measured (i.e. modulation of inflammatory neurotoxicity, cytokine/neurotransmitter compatibility, psychoneuroimmunology, etc…) and/or the biophilia hypothesis, or remove/adjust this otherwise superfluous information.
2. While there are few studies on forest bathing and adolescents, there are numerous studies on the impact of other nature-based interventions and green spaces in adolescent populations. It would be helpful to cite those, to increase credibility to your hypothesis that foerst bathing (specifically) is beneficial. For example:
a. Moll, A., Collado, S., Staats, H., & Corraliza, J. A. (2022). Restorative effects of exposure to nature on children and adolescents: A systematic review. Journal of Environmental Psychology, 84, 101884. https://doi.org/10.1016/J.JENVP.2022.101884
b. Zhang, Y., Mavoa, S., Zhao, J., Raphael, D., & Smith, M. (2020, September 2). The association between green space and adolescents mental well-being: A systematic review. International Journal of Environmental Research and Public Health. MDPI AG. https://doi.org/10.3390/ijerph17186640
3. Social media use and decreased interpersonal interactions are overlapping topics, but they are distinct and separate contributors to adolescent mental health issues. Recommend providing a citation for each of these separately.
4. The work of Clifford/Page and ANFT are just one organization promoting Forest Therapy in North America. They are the most well-known but they are not the only organization doing this work. It would be appropriate to state that this is just one approach, not THE approach, to North American forest therapy.
5. Are the different stages of ANFT FT relevant to the data (e.g. is it being explored which phase affects wellbeing the most? If not, than how is detailed description of each section of the standard sequence relevant? This is explored in the Intro Section 1.2 and also the Discussion, it does not seem necessary to have such a detailed description in both places. Also are any of the authors ANFT trained forest therapy guides or currently in training? If so that needs to be stated as a potential source of bias. ANFT is a commercial venture, and advocating for it’s methods specifically (rather than Forest Therapy in general) could be seen as a conflict of interest. Recommend shortening this section to give a general summary of the ANFT forest therapy experience.
6. The content regarding the “more than human world” as written is not relevant to the current study. Either expand/rationalize the inclusion of this material or eliminate.
7. Relocation of the third site to a “ocean beach with a grassy and shrubby open foredune environment” does not conform to the concept of “Forest bathing”. It is this reviewer’s understanding this is partially why ANFT refers to “Nature and Forest Therapy” rather than “Forest Bathing”. This should be addressed to avoid confusion to readers wondering how time at the beach could be considered “Forest Bathing”. IT is mentioned in section 4.3 but could be mentioned sooner to avoid confusion
8. It would be beneficial to cite examples of WEMWBS being used to measure well-being specifically regarding nature/greenspace exposures (e.g. Wang, R., Browning, M. H. E. M., Kee, F., & Hunter, R. F. (2023). Exploring mechanistic pathways linking urban green and blue space to mental wellbeing before and after urban regeneration of a greenway: Evidence from the Connswater Community Greenway, Belfast, UK. Landscape and Urban Planning, 235, 104739. https://doi.org/10.1016/J.LANDURBPLAN.2023.104739)
9. Procedure section 2.5 is thorough though unnecessarily repetitive in places (e.g. No need to describe photovoice process in Section 2.4 and then again separately for each walk. Recommend condensing this section accordingly.
10. Qualitative Analysis & Findings. There appears to be a discrepancy between the four themes (connection, noticing, gratitude, and care) mentioned In Section 3.3.and the sections in Findings (Relaxation and Peace, Mindful Qualities, Being Away). It is not clear why these are not congruent, the mismatch is confusing. Recommend aligning sections in 3.4 with the four themes mentioned, or otherwise explain why these four themes are NOT the subsections of 3.4
11. “Being Away” as a concept clearly taken from Kaplan’s ART and should be acknowledged so that unaware readers do not think this is a novel term being coined by the current study. In addition, there are unexplored opportunities in the Discussion section to link this theme discovered in your data directly with one of the cornerstone theories in the field
12. Discussion. There is a big difference between “significantly increases” and “could significantly increase”. The latter is an appropriate interpretation of your data. The former is unfounded unscientific speculation. For example, there is no control group and therefore as a non-experimental study cannot determine causation. Similarly it cannot be said the 3 visits “increased participant mean mental well-being by two points” without a comparative control group.
13. Similarly, Section 4.2 specifically mentions the importance of intention. While this is beneficial therapeutically, from a scientific perspective this is the definition of the placebo effect, and demonstrates it may not be the forest therapy experience at all that is producing the measured results. Would a second (control) group that stated similar positive intentions but didn’t actually participate in forest therapy have similar results? It is unknown because such a group wasn’t used and so data is not available to compare.
14. Limitations (would be benefitted by listing separately):
a. No control group
b. Students completed WEMWBS the day before first FB session. If responses can change from one FB session, they could certainly have changed between the time WEMWBS was completed and the following day before the start of FB. Thus these are not truly accurate pre-post FB WEMWBS scores, and should be stated as such. Similarly, pre-post WEMWBS data collection between the pre- (again, the day before FB 1 visit) and post (FB 3 visit) covers an extensive amount of time, and cannot be ascribed to solely the forest bathing experience (many other things occur in the life of an adolescent during a 3 week period). This should be mentioned as a limitation.
c. Participants’ are author’s students. This is listed in Section 4.4. but is relevant as a limitation of the current paper and not just something to consider for future research.

Reviewer 2 Report
Comments and Suggestions for Authors
88 - cite Pasanen et al.
111 - cite covid relief cares act
124 - and throughout - ensure you say mental well-being, rather than just well-being
187-188 - the one and 3 times is confusing - clear this up
205 - say a bit of what the pilot was
Justify or address the attrition of 24 to 16 - that is 33%
249-274 - lengthy with no added value
276 - again mental well-being
297-299 - Depression was not an aim of your work, I would not included this and you did not discuss this previously
300- Remove all personal pronouns - writing tone should be impersonal and objective - correct throughout
306-316- lengthy, lacks scientific writing style
317-325 - shorten, clarity needed, don't use examples of what you did not use/do
347 - 350 - see above comments
353 - time 1 or 2 or 3?
Specifics of google doc, google slides, etc is not needed. Page 9 has a lot of repeated information describing the specific spaces
I suggest putting the prompts in a table and referring to it - they are listed and repeated in multiple spaces - quite redundant
410-416 - repeated info again
435 - mental well-being
446 - need in a table
449 - Depression is not in your AIMS, 3 participants is not enough to demonstrate improvement
A table would be beneficial to report qualitative data.
4. discussion - this should not be a restatement of your results but rather specifically the results are the same or different from previous work and adds to the body of research
614 - mental
645 - you mention wellness and health 3 times in the paper - this is not the same as wellbeing or mental wellbeing. You are bringing in a separate concept here - I would remove
697- 699 - I would removed depression as it is a medical dx and this cannot be stated from a self-reported questionnaire, the quote here is also not related to depression
Thank you for your paper - I am excited to see your corrections
Comments on the Quality of English LanguageCorrect/remove all personal pronouns
